# Obligate coupling of CFTR pore opening to tight nucleotide-binding domain dimerization

**Csaba Mihályi[1,2], Beáta Töröcsik[1,2], László Csanády[1,2]***

[1]Department of Medical Biochemistry, Semmelweis University, Budapest, Hungary; [2]MTA-SE Ion Channel Research Group, Semmelweis University, Budapest, Hungary

**Abstract** In CFTR, the chloride channel mutated in cystic fibrosis (CF) patients, ATP-binding-induced dimerization of two cytosolic nucleotide binding domains (NBDs) opens the pore, and dimer disruption following ATP hydrolysis closes it. Spontaneous openings without ATP are rare in wild-type CFTR, but in certain CF mutants constitute the only gating mechanism, stimulated by ivacaftor, a clinically approved CFTR potentiator. The molecular motions underlying spontaneous gating are unclear. Here we correlate energetic coupling between residues across the dimer interface with spontaneous pore opening/closure in single CFTR channels. We show that spontaneous openings are also strictly coupled to NBD dimerization, which may therefore occur even without ATP. Coordinated NBD/pore movements are therefore intrinsic to CFTR: ATP alters the stability, but not the fundamental structural architecture, of open- and closed-pore conformations. This explains correlated effects of phosphorylation, mutations, and drugs on ATP-driven and spontaneous activity, providing insights for understanding CF mutation and drug mechanisms.

*For correspondence: csanady. laszlo@med.semmelweis-univ.hu

**Competing interests:** The authors declare that no competing interests exist.

## Introduction

The cystic fibrosis transmembrane conductance regulator (CFTR) is a chloride ion channel which belongs to the superfamily of ATP binding cassette (ABC) proteins (*Riordan et al., 1989*). Following phosphorylation of the cytosolic regulatory (R) domain by cAMP-dependent protein kinase (PKA), CFTR channels are gated by cytosolic ATP (*Figure 1a*). The ion pore within the TMDs opens upon formation of a tight head-to-tail NBD dimer stabilized by two molecules of ATP occluded at the dimer interface (*Vergani et al., 2005*); each nucleotide is sandwiched between the conserved Walker A and B sequences of one NBD and the signature sequence of the other (*Smith et al., 2002*; *Chen et al., 2003*). Pore closure accompanies NBD dimer disruption prompted by ATP hydrolysis at the composite interfacial site harbouring the Walker motifs of NBD2 (site 2) (*Vergani et al., 2005*): the other site (site 1) is catalytically inactive retaining ATP bound throughout multiple gating cycles (*Aleksandrov et al., 2002*; *Basso et al., 2003*). As evidence of such strict coupling between NBD and transmembrane domain (TMD) movements, a hydrogen bond between the side chains of NBD2 Walker-A threonine (T1246) and an arginine (R555) adjacent to the NBD1 signature sequence was shown to form in open, but not in closed, channels (*Vergani et al., 2005*). These two positions have co-evolved as a pair (*Vergani et al., 2005*), and in crystal structures of nucleotide-bound ABC NBD dimers the analogous side chains also form a hydrogen bond, the arginine serving as hydrogen donor and the threonine/serine side chain as acceptor (*Smith et al., 2002*; *Chen et al., 2003*). Furthermore, by disturbing proper spacing between the donor and acceptor sites, both shortening of the R555 and lengthening of the T1246 side chains (by single mutations R555K and T1246N,

**eLife digest** A protein pore called the CFTR channel allows chloride ions to move through the membrane of the cells that line the airways and some other parts of the human body. Mutations in the genes that encode CFTR may reduce the number of pores at the cell surface or stop them from working properly. When this happens, these cells cannot transport enough chloride, which causes the disease cystic fibrosis. CFTR contains two regions that lie inside the cell known as nucleotide binding domains (NBDs). These domains bind to the chemical energy molecule ATP, and when ATP does bind, two NBDs associate to form a dimer and the pore in CFTR opens.

The CFTR channel can occasionally open in a spontaneous way that does not require ATP. However, it was not clear whether NBDs also formed dimers when CFTR opened in this way. This is because spontaneous opening could reflect NBDs occasionally forming a dimer without ATP binding or it could occur when the pore occasionally opens without the NBDs forming a dimer.

To explore whether opening of the pore always requires NBD dimerization, Mihályi et al. studied the behaviour of single human CFTR channels produced in frog eggs. Normal channels and mutant ones (which show differences in spontaneous opening) were used, and the change in the way NBDs interacted when the channels spontaneously opened or closed was investigated. Mihályi et al. found that the NBD dimer forms when the pore spontaneously opens, demonstrating that this step happens both with and without ATP.

The result demonstrates that NBD dimer formation and pore movement are strictly coupled and that this is an inbuilt property of the CFTR protein. When ATP binds, this only changes how stable the open-pore and closed-pore structures of CFTR are but does not alter the fundamental architecture of the channel.

These new findings will be of interest to researchers studying a large group of transport proteins related to CFTR called ABC proteins. Furthermore, a drug called ivacaftor stimulates spontaneous opening of CFTR, and has recently been approved for clinical use to treat people with mutations in CFTR. As such, the new findings will be also useful to help researchers understand how ivacaftor stimulates the CFTR pore to open.

respectively) prevent, whereas the double mutation R555K-T1246N restores, the formation of this stabilizing hydrogen bond (*Vergani et al., 2005*).

Wild-type (WT) CFTR channels open occasionally even in the absence of ATP (*Szollosi et al., 2010*) (*Figure 1a*, inset), but it is unclear whether this reflects the occasional spontaneous formation of the NBD dimer (*Figure 1b*, top), or occasional pore openings independent of NBD dimerization (*Figure 1b*, bottom). The significance of this question goes beyond the scarcity of WT spontaneous openings. Indeed, the strictness of coupling between NBD and TMD movements is a strongly debated fundamental question, key for understanding the forces that drive the normal, ATP-dependent functional cycle of ABC proteins (*Vergani et al., 2005*; *Csanády et al., 2010*; *Kirk and Wang, 2011*; *Jih and Hwang, 2013*; *Okeyo et al., 2013*). Moreover, ivacaftor, the only potentiator drug to be approved so far for clinical use in CF patients, acts by promoting such spontaneous, ATP-independent, openings of G551D CFTR, the third most common disease mutant (*Bompadre et al., 2007*; *Jih et al., 2013*). However, mechanistic understanding of spontaneous gating has been limited by its vanishingly small open probability ($P_o$), far too low to be reliably quantified (*Figure 1a*). Recently, mutations in the third intracellular loop (position 978, between TM helices 8 and 9) (*Wang et al., 2010*), and at the cytosolic end of pore-lining TM helix 6 (position 355) (*Wei et al., 2014*), were both found to strongly enhance spontaneous activity. Furthermore, these 'gain-of-function' effects proved additive in the double-mutant P355A–K978C, shifting spontaneous $P_o$ into a range where quantitative biophysical studies become tractable (*Wei et al., 2014*). Here we exploit this double mutant as a background to address the mechanism of spontaneous openings.

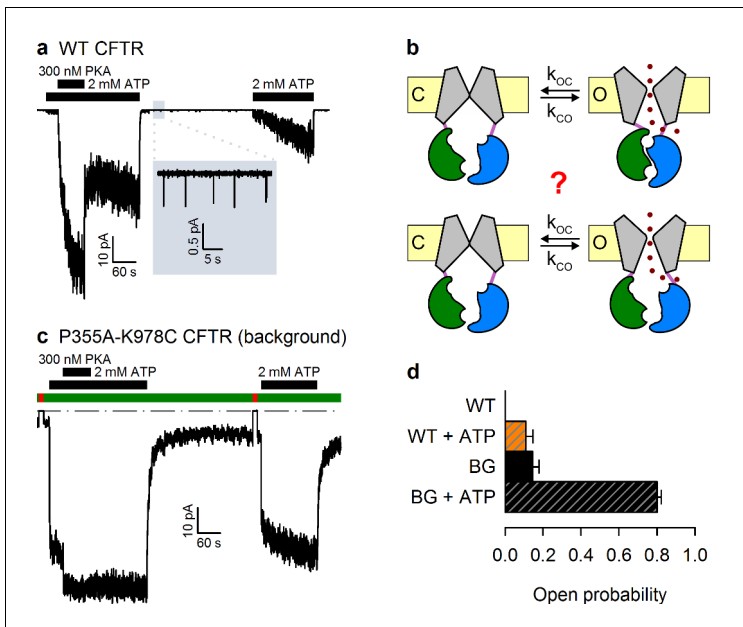

**Figure 1.** The P355A-K978C double mutation increases both spontaneous and ATP-dependent $P_o$. (a,c) Inward macroscopic (a) WT and (c) P355A-K978C CFTR currents at −80 mV, and their dependence on ATP and PKA (*black bars*). In (a) a 30 s segment of spontaneous current (*gray box*) is magnified in the *inset*. In (c) bath chloride (*green bars*) was repeatedly replaced by gluconate (*red bars*) to determine baseline current (*gray dashed line*). (b) Two alternative mechanisms of spontaneous gating in cartoon representation. TMDs (*gray*), TMD-NBD interface formed by intracellular loops (*light violet*), NBD1 (*green*), NBD2 (*blue*), membrane (*yellow*). (d) Spontaneous $P_o$ (*solid bars*) and $P_o$ in 2 mM ATP (*striped bars*) for WT and P355A-K978C ('background', BG) CFTR ($n$ = 5–21).

The following figure supplements are available for figure 1:

**Figure supplement 1.** Replacement of bath chloride with gluconate abolishes CFTR currents at -80 mV.

**Figure supplement 2.** High ATP-dependent $P_o$ due to background mutation facilitates counting channels.

# Results

## Gain-of-function mutations allow quantitative characterization of spontaneous CFTR gating

Whereas WT CFTR channel currents decline rapidly to baseline upon ATP removal (*Figure 1a*), in patches containing P355A-K978C CFTR channels (*Figure 1c*) a readily measurable fraction of the chloride current resisted even prolonged removal of bath ATP, but could be abolished by removal of permeating chloride ions (*red bars*; see also *Figure 1—figure supplement 1*). Moreover, for WT CFTR, ATP-dependent currents are very small prior to exposure to PKA, and decline by ~50% when PKA is withdrawn (*Figure 1a*), signalling strong phosphorylation dependence of its $P_o$ (*Szollosi et al., 2010*). In contrast, ATP-evoked P355A-K978C CFTR currents were large already before exposure to PKA, and were little further enhanced by the addition of – or reduced following removal of – the kinase (*Figure 1c*), suggesting a large $P_o$ for unphosphorylated (pre-PKA) double-mutant channels in ATP, and close-to-maximal $P_o$ when the channels were either fully phosphorylated (in PKA) or partially dephosphorylated (post-PKA). The robust spontaneous activity of P355A-K978C channels in the absence of ATP allowed reliable sampling of the characteristics of spontaneous gating in microscopic patches (*Figure 2a*), in which the number of active channels could be confidently estimated from bracketing high-$P_o$ segments of recordings in ATP (*Figure 1—figure supplement 2*). Thus, for pre-phosphorylated channels the P355A-K978C background mutation increased spontaneous $P_o$ from $0.000053 \pm 0.000021$ ($n$ = 5) to $0.15 \pm 0.03$ ($n$ = 21) (*Figure 1d*, WT vs. BG [background]), and $P_o$ in ATP from $0.11 \pm 0.04$ ($n$ = 6) to $0.80 \pm 0.02$ ($n$ = 14) (*Figure 1d*, *striped bars*).

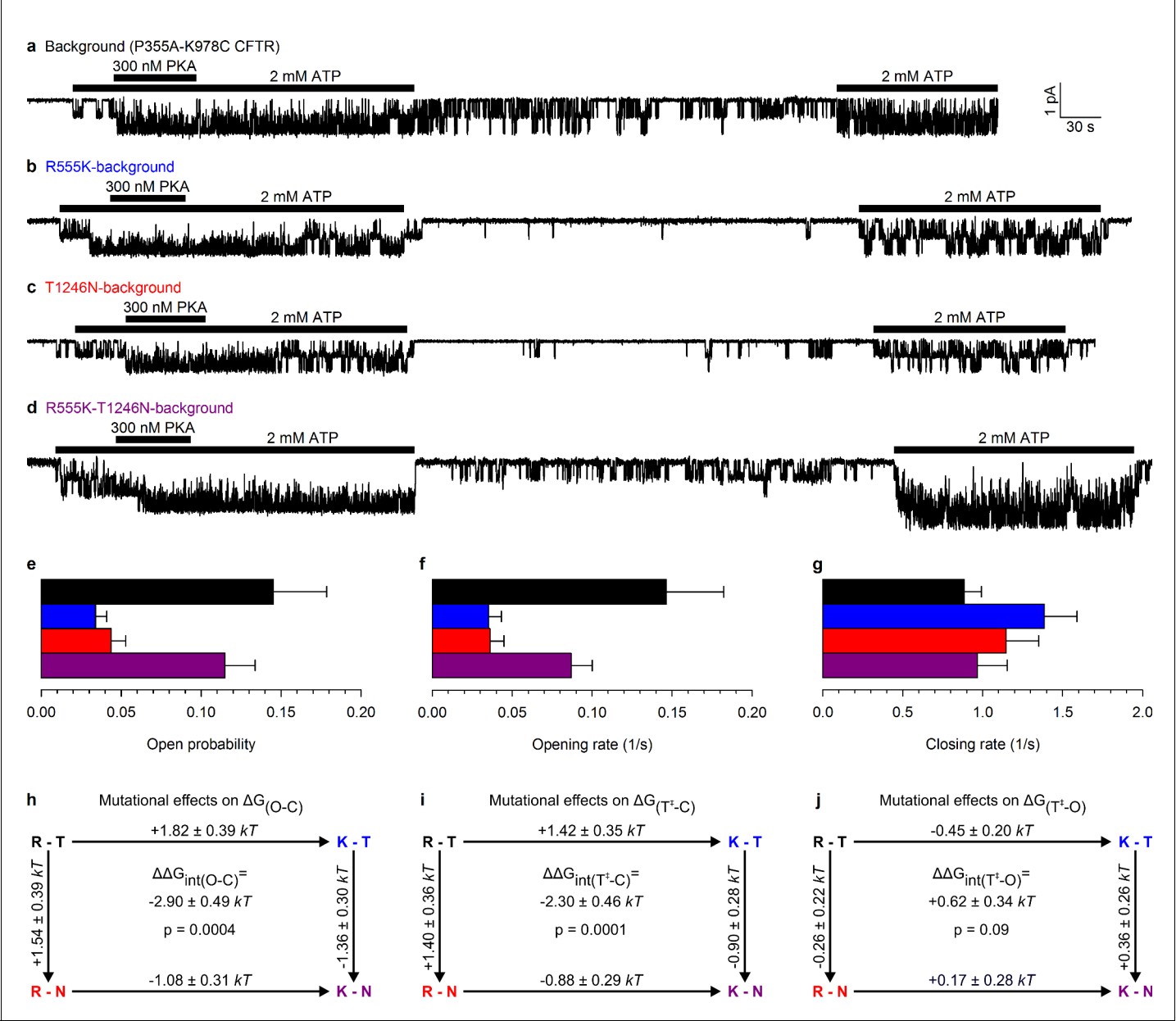

**Figure 2.** Arg 555 and Thr 1246 become energetically coupled upon spontaneous pore opening. (**a–d**), Microscopic currents at −80 mV, and their dependence on ATP and PKA (*black bars*), for (**a**) the P355A-K978C background construct, and for this background construct carrying (**b**) the R555K or (**c**) T1246N single mutations or (**d**) the R555K-T1246N double mutation. (**e–g**), Spontaneous open probabilities (**e**, *n* = 19–21), opening rates (**f**, *n* = 15–19), and closing rates (**g**, *n* = 15–19) of the background construct (*black bars*), and of constructs carrying additional R555K (*blue bars*), T1246N (*red bars*), or R555K-T1246N (*violet bars*) mutations. (**h–j**), Thermodynamic mutant cycles showing mutation-induced changes (**h**) in the stability of the spontaneous open state relative to the closed state, and (**i–j**) in the height of the activation free energy barriers for spontaneous opening (**i**) and closure (**j**). Each corner is represented by the side chains at positions 555 and 1246; *k*, Boltzmann's constant, *T*, absolute temperature.

## The site-2 NBD interface is tightly dimerized in the spontaneous open-channel state

This double-mutant CFTR background allowed us to study, using the thermodynamic double mutant cycle formalism (Materials and methods), changes in energetic coupling associated with spontaneous gating between the same two NBD-interface side chains previously used (*Vergani et al., 2005*) to demonstrate NBD dimer formation and disruption in CFTR channels opening and closing in ATP.

Mutating either the arginine at position 555 to a lysine, or the threonine at position 1246 to an asparagine, resulted in a large reduction in spontaneous $P_o$ (*Figure 2b–c* 0.034 ± 0.007 (*n* = 18) and 0.044 ± 0.009 (*n* = 19), respectively, *Figure 2e*, *blue and red bar*), that is, in the closed-open equilibrium constant in the absence of ATP. This reduction in the stability of the open (O) state relative to the closed (C) state ( $\Delta\Delta G^\circ_{(O-C)}$ = +1.82 ± 0.39 *kT* and +1.54 ± 0.39 *kT*, respectively, in R555K and T1246N single mutants; *Figure 2h*, numbers on top-across, and left-downward, arrows of the mutant cycle) is due to the loss in both single mutants of microscopic interactions, between the targeted side chain and the rest of the protein, that in WT channels combine to increase the stability of the open state.

If the side chains of residues R555 and T1246 did not interact in WT CFTR, or if their interaction were to remain unaltered during spontaneous gating, then any impact on the spontaneous channel activity of mutating either single position should be independent of the nature of the side chain found at the other position: that is, mutational effects at the two positions should add energetically. Any non-additive behaviour would signal gating-associated dynamic changes in the interaction energy between the R555 and T1246 side chains. In stark contrast to the single mutants, the double mutant R555K-T1246N (*Figure 2d*) displayed a high spontaneous $P_o$ (0.11 ± 0.02 (*n* = 20); *Figure 2e*, *violet bar*), comparable to that of the background construct. Thus, the effects on the spontaneous $P_o$ of either single mutation depended strongly on the nature of the residue at the other position: the spontaneous open-channel state was destabilized relative to the closed-channel state when the R555K mutation was introduced into a WT background (*blue* vs. *black bar* in *Figure 2e*), but was stabilized when the same mutation was introduced into a T1246N background (*violet* vs. *red bar* in *Figure 2e*). The difference between $\Delta\Delta G^\circ_{(O-C)}$ along two parallel sides of the mutant cycle quantifies the change in interaction strength between the two target side chains that accompanies a spontaneous pore opening ($\Delta\Delta G_{int(O-C)}$; see Materials and methods), and revealed a significant (p = 0.0004) change of –2.90 ± 0.49 *kT* (*Figure 2h*, *center*) in R555-T1246 interaction energy between ATP-free closed- and open-channel states. In principle, the negative sign of $\Delta\Delta G_{int(O-C)}$ is compatible with the existence of either a stabilizing interaction in the spontaneously formed open-channel state, or a destabilizing interaction in the closed state. However, earlier work reported a lack of interaction between the R555 and T1246 side chains in closed channels, regardless of the presence of ATP in site 2 (*Vergani et al., 2005*). Thus, the only plausible explanation is a stabilizing interaction, consistent with the formation of a hydrogen bond, in the spontaneous open state in WT channels that is lost in both single mutants, but is restored in the double mutant. The inevitable implication of these findings is that the NBD dimer interface must tighten within site 2 upon pore opening, whether or not bound ATP is present.

## The site-2 NBD interface is already tightened in the transition state for spontaneous pore opening

Conformational transitions between stable open-channel (burst) and closed-channel (interburst) states of CFTR reflect passages across the transition ($T^\ddagger$) state (*Figure 3a*), a high free-energy transient conformation. The rates of spontaneous opening and closure of each molecular channel type, obtained as the inverse of the mean interburst ($\tau_{ib}$) and burst ($\tau_b$) durations, respectively, report on transition-state stability: mutation-induced fractional changes in spontaneous opening and closing rates quantitate changes in transition-state free energy relative to the closed ($\Delta\Delta G^\circ_{(T-C)}$) and open ($\Delta\Delta G^\circ_{(T-O)}$) ground states, respectively. Specifically, the difference between such $\Delta\Delta G^\circ$ values along two parallel sides of a mutant cycle quantify the changes in coupling strength between the two target side chains upon reaching the transition state during opening ($\Delta\Delta G_{int(T-C)}$, *Figure 2i*) or closure ($\Delta\Delta G_{int(T-O)}$, *Figure 2j*). Kinetic analysis revealed that the reduced spontaneous open probabilities of both single mutants R555K and T1246N (*Figure 2e*, *blue* and *red bar*) were caused predominantly by a markedly reduced opening rate (*Figure 2f*, *blue* and *red bar*), which was restored in the double mutant (*Figure 2f*, *violet bar*). Consequently, a significant (p = 0.0001) negative value of –2.30 ± 0.46 *kT* (*Figure 2i*, *center*) was obtained for the change in coupling free energy between the target side chains upon reaching the transition state from the closed state ($\Delta\Delta G_{int(T-C)}$). In contrast, closing rates were little affected by the mutations (*Figure 2g*), yielding a value of +0.62 ± 0.34 *kT* for $\Delta\Delta G_{int(T-O)}$, not significantly different from zero (*Figure 2j*, *center*). These results imply that in WT

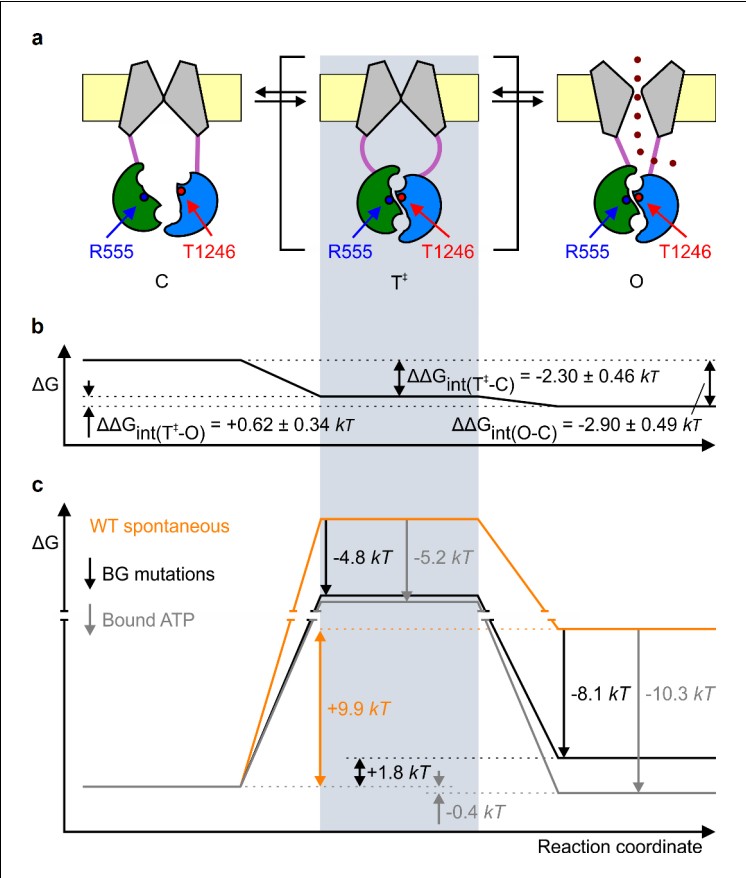

**Figure 3.** Mechanism of spontaneous pore openings, and energetic effects of bound ATP and gain-of-function mutations. (a) Cartoon representation of domain organization in closed (C), transition (T‡), and open (O) states, during spontaneous gating. Color coding as in *Figure 1b*, *blue* and *red circles* identify target positions. (b) Changes in energetic coupling between the target positions associated with spontaneous gating in the background construct. (c) Free energy profiles of gating for WT (*orange line*) and P355A-K978C (*black line*) CFTR in the absence of ATP, and of hydrolysis-deficient K1250R CFTR in saturating ATP (*gray line*) (*Vergani et al., 2005*). *Black* and *gray downward arrows* illustrate the energetic effects of the background double mutation and of the presence of ATP bound in both composite sites.

channels the hydrogen bond between the R555 and T1246 side chains is already formed in the transition state for spontaneous opening, its strength being little further altered when the protein reaches the stable open state (*Figure 3b*). Thus, the NBD dimer interface tightens within site 2 already in the transition state for unliganded opening (*Figure 3a*, *center*), just as it does during the opening of ATP-bound channels (*Vergani et al., 2005*; *Sorum et al., 2015*).

## Discussion

While ATP-driven CFTR pore openings are known to be coupled to the formation of a tight NBD dimer (*Vergani et al., 2005*), the mechanism of spontaneous openings, observable in the absence of ATP, is unknown. Here we show that spontaneous openings are also linked to tightening of the NBD interface around composite site 2, just as in the presence of bound ATP. Such strict coupling between NBD and pore movements – regardless of the absence or presence of ATP – strongly constrains mechanistic models of ATP-dependent channel gating, and understanding of the underlying driving forces, which are likely conserved among ABC proteins. It also helps explain modulatory effects of mutations, or of physiological or pharmacological gating regulators. For example, the G551D CF mutation converts ATP at site 2 into an inhibitory ligand, due to electrostatic repulsion

between the γ-phosphate of ATP bound to the Walker motifs, and the introduced negative charge in the signature sequence (*Lin et al., 2014*). Such an effect would not be expected if the opposing NBD interfaces did not approach each other in the open-channel state, but is readily explained by openings being strictly linked to NBD dimer formation even in the G551D mutant. Furthermore, although reported low-probability ATP-independent pore openings of a CFTR construct lacking NBD2 (CFTR residues 1–1197; [*Wang et al., 2007*]) can clearly not be linked to NBD dimer formation, the present results argue that upon pore opening its remaining NBD-TMD coupling machinery undergoes movements similar to those that accompany NBD dimerization when neither 'limb' is truncated.

Because of the tightened NBD interface, the ATP binding sites of open CFTR channels are likely inaccessible to the bulk solution even in the absence of bound ATP, just as in its presence (*Chaves and Gadsby, 2015*), suggesting that ATP binding/unbinding can take place only in the closed-channel state. Two immediate corollaries ensue. First, the ATP-free open-channel state is unlikely to be an integral component of the ATP-dependent gating cycle (*Szollosi et al., 2010*; but, cf. *Jih et al., 2013*). Second, despite suggested similarities (*Wang et al., 2010*; *Kirk et al., 2011*; *Okeyo et al., 2013*), the principles that drive gating of CFTR channels are fundamentally different from those of ligand-gated channels, such as the nicotinic acetylcholine receptor. Ligand-gated channels obey cyclic equilibrium mechanisms in which ligand binding/unbinding can occur both in the closed- and the open-channel state, and the activating effect of the ligand is based on the thermodynamic principle of detailed balance which constrains the product of the equilibrium constants around a cycle to be unity: thus, opening ('isomerization') is driven by the concurrent conformational change of the ligand binding site which increases its affinity for the ligand (*Grosman and Auerbach, 2001*). But in CFTR the ligand binding site is occluded in open channels, so that an open-state 'binding affinity' cannot be defined (the rates for both ATP binding and unbinding are zero). Thus, even when disruption of ATP hydrolysis forces CFTR channels to gate at equilibrium, that gating follows a reversible linear sequence (cf., *Figure 3a*), rather than a cycle subject to the principle of detailed balance.

Strong correlation between effects on unliganded and liganded channel activity, and energetic additivity of such effects, has been reported for a range of diverse 'allosteric modulators' acting on various regions of the channel. For instance, both unliganded and liganded $P_o$ is increased in a physiological context by R-domain phosphorylation, presumably by altering R-domain interactions with the TMDs (*Wang et al., 2010*), or in a therapeutic context by the CF drug ivacaftor, believed to target the TMDs (*Jih et al., 2013*). Further, distinct mutations in intracellular loops increase (*Wang et al., 2010*; *Wei et al., 2014*) or decrease (*Wang et al., 2014*) $P_o$ for both unliganded and liganded CFTR. The present results suggest a simple, unified explanation for all these findings: energetic stabilization or destabilization of the inherent open-channel-state structure (*Figure 3a*, *right*), which we show here is similarly organized in the absence or presence of ATP.

Comparison of gating parameters for WT CFTR and our background mutant reveals the mechanism by which gain-of-function mutations promote spontaneous activity (*Figure 3c*, *black* vs. *orange* free energy profiles). The increase in spontaneous $P_o$ by the P355A-K978C double mutation (*Figure 1d*, BG vs. WT) amounts to a stabilization of the spontaneous open state by ~8.1 $kT$ (*Figure 3c*, *right black arrow*), and is due not only to an increased frequency of openings but also to a slower rate of closure, because spontaneous $\tau_b$ is substantially prolonged (from $60 \pm 9$ ms ($n = 5$) to $1566 \pm 235$ ms ($n = 19$)), suggesting an increase in $\Delta\Delta G^\circ_{(T-O)}$ of ~3.3 $kT$ relative to WT. Together, these data imply that these gain-of-function mutations stabilize the transition state by far less (only by ~4.8 $kT$; *Figure 3c*, *left black arrow*) than the open state, as expected for perturbations that affect protein positions which have not yet reached their fully open-like conformations in the transition state. This is consistent with a similar finding for an NBD-TMD interface position during ATP-dependent opening (*Sorum et al., 2015*). Thus, just as for ATP-bound channels, the pore is likely still closed in the high-energy evanescent transition state for spontaneous openings (*Figure 3a*, *center cartoon*).

Finally, what can we infer about the mechanism by which binding of the natural ligand ATP promotes CFTR channel opening? The presence of bound ATP does not alter the global arrangement of conformations: rather ATP acts as a 'molecular glue' to stabilize the transition state and the open state. Because ATP-induced openings are terminated by ATP hydrolysis (*Csanády et al., 2010*),

energetic interpretation of ATP-binding induced gating effects cannot be based on the comparison of WT gating in the presence and absence of ATP (*Figure 1d*, WT+ATP vs. WT). However, comparing gating parameters for spontaneous activity of WT CFTR and ATP-driven activity of hydrolysis-incompetent mutant CFTR channels offers some insight into the impact of ATP binding on gating energetics (*Figure 3c*, *gray* vs. *orange* free energy profiles). Mutations that disrupt ATP hydrolysis in site 2 all prolong $\tau_b$ and reduce CFTR gating to an equilibrium, by forcing the channels to close back through the same high-energy transition state they had traversed upon opening (cf. *Figure 3a*) (*Vergani et al., 2005*; *Csanády et al., 2010*; *Sorum et al., 2015*). For instance, comparison of $P_o$ and $\tau_b$ of the site 2 Walker-A mutant K1250R CFTR in saturating ATP ($P_o \sim 0.6$, $\tau_b = 9323$ ms [*Vergani et al., 2005*]) with those of spontaneously gating WT CFTR suggests that the presence of bound ATP stabilizes the open-channel state by ~10.3 $kT$ (*Figure 3c*, *right gray arrow*) and increases $\Delta\Delta G^{\circ}_{(T-O)}$ by ~5.1 $kT$, respectively, which implies that bound ATP stabilizes the transition state by only ~5.2 $kT$ (*Figure 3c*, *left gray arrow*). Thus, intriguingly, although the NBD interface is already tightened within site 2 in the ATP-bound transition state (*Vergani et al., 2005*; *Sorum et al., 2015*), the 'molecular glue' effect of bound ATP is further strengthened when the channel reaches the stable open state. The implication is that some rearrangements must still occur within the dimer interface between the transition state and the open state. One possibility is that, in the transition state, site 1 has not yet reached its full open-channel-like conformation. Perhaps the site 2 'glue' bonds fully already in the transition state, whereas bonding of the site 1 'glue' is completed only in the open-channel state. Further work will be needed to address the precise timing of site 1 motions during channel opening, and their contributions to CFTR gating energetics.

## Materials and methods

### Molecular biology

Mutations introduced (Stratagene, QuikChange) into pGEMHE-CFTR were confirmed by automated sequencing; cDNA was transcribed in vitro (Ambion, mMessage T7) and cRNA stored at −80°C.

### Isolation and injection of Xenopus oocytes

Oocytes were isolated from anaesthetized adult female *Xenopus* laevis frogs following Institutional Animal Care Committee guidelines, injected with 0.1–10 ng cRNA in a fixed 50 nl volume, and stored at 18°C as described (*Szollosi et al., 2010*). Current recordings were obtained 1–3 days after injection.

### Excised inside-out patch recording

The patch pipette solution contained (in mM): 136 NMDG-Cl, 2 $MgCl_2$, 5 HEPES, pH = 7.4 with NMDG. Bath solution contained (in mM): 134 NMDG-Cl, 2 $MgCl_2$, 5 HEPES, 0.5 EGTA, pH = 7.1 with NMDG, and was freshly supplemented with 3 mM DTT to prevent thiol oxidation of the cysteine side chain engineered into position 978. MgATP (2 mM) was added from a 400 mM aqueous stock solution (pH = 7.1 with NMDG). The catalytic subunit of PKA (300 nM, Sigma) was applied for ~1 min to activate CFTR channels. Experiments were done at 25°C. Currents were recorded at a membrane potential of −80 mV, digitized at 10 kHz, and recorded to disk after on-line Gaussian filtering at 2 kHz.

For macroscopic recordings of P355A-K978C CFTR, baseline current was estimated by brief replacements of bath chloride ions with gluconate (*Figure 1c*, *red bars*); at −80 mV no CFTR currents are observable under such ionic conditions (*Figure 1—figure supplement 1*). For microscopic recordings channels were pre-phosphorylated by PKA, and gating in the absence of ATP was observed for 5 min, bracketed by 2.5 min exposures to ATP. The number of active channels was estimated based on the high-$P_o$ bracketing segments (cf., *Figure 1—figure supplement 2*). To allow for loss of ATP even from the higher affinity site 1, which retains ATP for tens of seconds (*Tsai et al., 2010*), spontaneous gating parameters were extracted from the last ~4 min stretches of the 5 min ATP-free segments; precise starting points for the quasi-steady segment to be analyzed were chosen in each patch based on visual inspection.

## Kinetic analysis of microscopic patches

Segments of current recording were digitally filtered at 50 Hz, and idealized by half-amplitude threshold crossing. Open probabilities in the absence (*Figure 2e*) and presence (*Figure 1—figure supplement 2*) of ATP were calculated from the events lists as the time-average of the fraction of open channels. Spontaneous channel opening and closing rates were calculated as described (*Vergani et al., 2005*), from ATP-free segments containing no more than 7 superimposed channel openings. A closed-open-blocked (C-O-B) kinetic scheme, which separates brief (~10 ms) flickery closures (to state B) from long (>1 s) interburst closures (to state C), was fitted by maximum likelihood to the set of dwell-time histograms for all conductance levels, to obtain microscopic transition rates $r_{CO}$, $r_{OC}$, $r_{OB}$, and $r_{BO}$, while accounting for a fixed dead time of 6 ms (*Csanády, 2000*). Mean burst ($\tau_b$) and interburst ($\tau_{ib}$) durations were calculated as $\tau_b = (1/r_{OC})(1 + r_{OB}/r_{BO})$ and $\tau_{ib} = 1/r_{CO}$, and channel opening and closing rates defined as $1/\tau_{ib}$ and $1/\tau_b$, respectively.

## Mutant cycle analysis

Changes in interaction strength between pairs of residues associated with various steps of the gating cycle were estimated using mutant cycle analysis, as described previously (*Vergani et al., 2005*). For a conformational transition from state A to state B of the protein, energetic coupling ($\Delta\Delta G_{int}$) between positions 1 and 2 is defined as the difference between mutation-induced changes in the stability of state B relative to state A along parallel sides of the mutant cycle, i.e.,

$$\Delta\Delta G_{int} = \Delta\Delta G_{B-A}^{0(X'Y \to X'Y')} - \Delta\Delta G_{B-A}^{0(XY \to XY')} = \Delta\Delta G_{B-A}^{0(XY' \to X'Y')} - \Delta\Delta G_{B-A}^{0(XY \to X'Y)} \tag{1}$$

Here the superscripts refer to the pairs of residues found at target positions 1 and 2 of the four studied constructs (e.g., XY: residue X at position 1 and residue Y at position 2; XY': residue X at position 1 and residue Y' at position 2; etc.), and arrows represent substitutions in a particular background (e.g., XY→XY': mutation of position 2 from Y to Y', with X at position 1).

If states A and B are stable ('ground') states of the protein, then the energetic stability of state B relative to state A can be calculated from the measured equilibrium constant ($K_{eq}$) as $\Delta G_{B-A}^0 = -kT \ln K_{eq}$ ($k$ is Boltzmann's constant, $T$ is absolute temperature), and hence the two terms in *Equation 1* are obtained as follows:

$$\Delta\Delta G_{B-A}^{0(X'Y \to X'Y')} = \Delta G_{B-A}^{0(X'Y')} - \Delta G_{B-A}^{0(X'Y)} = -kT \ln\left(K_{eq}(X'Y')/K_{eq}(X'Y)\right),$$

and

$$\Delta\Delta G_{B-A}^{0(XY \to XY')} = \Delta G_{B-A}^{0(XY')} - \Delta G_{B-A}^{0(XY)} = -kT \ln\left(K_{eq}(XY')/K_{eq}(XY)\right).$$

In this study, we used the equilibrium constant between the spontaneous closed- and open-channel state, $K_{eq} = P_o/(1 - P_o)$ (*Figure 2h*).

Analogously, mutation-induced changes in activation free energy barriers, calculated from *fractional* changes in transition rates, can also be substituted into *Equation 1*. For instance, for the transition A→T$^\ddagger$→B (T$^\ddagger$ is the transition state) the two terms for *Equation 1* are obtained as follows:

$$\Delta\Delta G_{T-A}^{0(X'Y \to X'Y')} = -kT \ln(r_{AB}(X'Y')/r_{AB}(X'Y)),$$

and

$$\Delta\Delta G_{T-A}^{0(XY \to XY')} = -kT \ln(r_{AB}(XY')/r_{AB}(XY)),$$

where $r_{AB}$ is the transition rate from state A to state B. In this study, we calculated mutation-induced changes in activation free energy barriers for channel opening ($\Delta\Delta G_{T-C}^0$) and closing ($\Delta\Delta G_{T-O}^0$) from the fractional changes in opening and closing rates, respectively (*Figure 2i–j*). All $\Delta\Delta G$s are given as mean ± SEM. Estimates of SEM for $\Delta\Delta G^o$ values were obtained as follows. All $\Delta\Delta G$ values are functions of the form $\Delta\Delta G = kT \ln(u/v)$ of two independent random variables $u$ and $v$ which represent a specific gating parameter ($K_{eq}$, $r_{CO}$, or $r_{OC}$) of two different channel constructs. Assuming that $u$ and $v$ are normally distributed with mean m ($m_u, m_v$) and variance $\sigma^2$ ($\sigma_u^2, \sigma_v^2$), the variance of $\ln(u/v)$ is given by

$$\text{var}(\ln(u/v)) = \int_{-\infty}^{\infty}\int_{-\infty}^{\infty}\ln^2(u/v)\cdot\frac{1}{2\pi}\cdot e^{\frac{-(u-m_u)^2}{2\sigma_u^2}}\cdot e^{\frac{-(v-m_v)^2}{2\sigma_v^2}}\,dudv - \left(\int_{-\infty}^{\infty}\int_{-\infty}^{\infty}\ln(u/v)\cdot\frac{1}{2\pi}\cdot e^{\frac{-(u-m_u)^2}{2\sigma_u^2}}\cdot e^{\frac{-(v-m_v)^2}{2\sigma_v^2}}\,dudv\right)^2$$

The above integrals cannot be performed analytically, but the functions $ln(u/v)$ and $ln^2(u/v)$ can be expanded into power series around $(m_u, m_v)$. Taking the terms only up to second order, the integrals can be performed, giving the approximation

$$\text{var}(\ln(u/v)) \approx \frac{\sigma_u^2}{m_u^2} + \frac{\sigma_v^2}{m_v^2} + \frac{1}{2}\cdot\frac{\sigma_u^2}{m_u^2}\cdot\frac{\sigma_v^2}{m_v^2} - \frac{1}{4}\cdot\left(\frac{\sigma_u^4}{m_u^4} + \frac{\sigma_v^4}{m_v^4}\right)$$

The power series approximations hold as long as $\sigma_u \ll m_u$, $\sigma_v \ll m_v$; thus, for $\sigma_u = 0.2m_u, \sigma_v = 0.2m_v$ this estimate of the variance for $\ln(u/v)$ is still quite accurate, and was used to obtain $\text{var}(\Delta\Delta G_{B-A}^0)$ values. Since the numbers of observations for each pair of constructs (e.g., $n_{XY}$ and $n_{XY'}$) was similar, standard errors were calculated using the mean values for $n$. E.g., $\text{SEM}\left(\Delta\Delta G_{B-A}^{0\,(XY\to XY')}\right) = \sqrt{\text{var}\left(\Delta\Delta G_{B-A}^{0\,(XY\to XY')}\right)/\bar{n}}$, where $\bar{n} = (n_{XY} + n_{XY'})/2$.

Variances for energetic coupling were obtained as $\text{var}(\Delta\Delta G_{int}) = \text{var}\left(\Delta\Delta G_{B-A}^{0\,(X'Y\to X'Y')}\right) + \text{var}\left(\Delta\Delta G_{B-A}^{0\,(XY\to XY')}\right)$. Finally, since the numbers of observations for each corner of a cycle ($n_{XY}, n_{XY'}, n_{X'Y}, n_{X'Y'}$) were similar, standard errors for energetic coupling were calculated as $\text{SEM}(\Delta\Delta G_{int}) = \sqrt{\text{var}(\Delta\Delta G_{int})/\bar{n}}$, where $\bar{n} = (n_{XY} + n_{XY'} + n_{X'Y} + n_{X'Y'})/4$.

The values of $\Delta\Delta G_{int}$ obtained from the above two types of mutant cycle are conventionally interpreted to reflect the change in the strength of the interaction (in WT) between the two target residues X and Y upon transiting from stable ground state A either to stable ground state B, or to transition state $T^\ddagger$, respectively. This interpretation assumes that the native interaction between residues X and Y is destroyed in both single mutants and the double mutant (valid, e.g., for side chain truncations) (*Horovitz, 1996*). However, if an interaction is restored in the double mutant X'Y', then the measured $\Delta\Delta G_{int}$ reports the sum of the changes in interaction strengths for residue pairs X-Y (in WT) and X'-Y' (in the double mutant). Choosing such mutations (e.g., (*Vergani et al., 2005*); present study) provides a convenient signal amplification which facilitates separating the signal from noise.

## Statistics

All data are presented as mean ± S.E.M, estimated from current recordings obtained from $n$ ($\geq 5$) patches as specified in each figure legend. All data were included in the analysis. Statistical significance was quantified using Student's $t$ test, and differences are reported as significant for $p < 0.05$.

## Acknowledgements

Supported by MTA Lendület grant LP2012-39/2012, Cystic Fibrosis Foundation Research Grant CSANAD15G0, and Howard Hughes Medical Institute International Early Career Scientist grant 55007416 to LC.

## Additional information

### Funding

| Funder | Grant reference number | Author |
|---|---|---|
| Howard Hughes Medical Institute | International Early Career Scientist Grant 55007416 | László Csanády |
| Magyar Tudományos Akadémia | Lendület grant LP2012-39/2012 | László Csanády |
| Cystic Fibrosis Foundation | Research Grant CSANAD15G0 | László Csanády |

The funders had no role in study design, data collection and interpretation, or the decision to submit the work for publication.

## Author contributions

CM, Performed all experiments, Analyzed the data, Drafting or revising the article; BT, Generated all mutant constructs; LC, Designed the project, Assisted in data analysis, Wrote the manuscript

## Author ORCIDs

Csaba Mihályi, http://orcid.org/0000-0001-7536-3066
László Csanády, http://orcid.org/0000-0002-6547-5889

## Ethics

Animal experimentation: This study was performed in strict accordance with the recommendations in the Guide for the Care and Use of Laboratory Animals of the National Institutes of Health. All of the animals were handled according to approved institutional animal care and use committee (IACUC) protocols of Semmelweis University (22.1/1935/3/2011).

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
