## [Decision Letter]

Thank you for submitting your article "Obligate coupling of CFTR pore opening to tight nucleotide-binding domain dimerization" for consideration by *eLife*. Your article has been favorably evaluated by Richard Aldrich (Senior editor) and three reviewers, one of whom, Kenton J Swartz (Reviewer #1), is a member of our Board of Reviewing Editors. The following individual involved in review of your submission has agreed to reveal their identity: Christopher Miller (Reviewer #3).

The reviewers have discussed the reviews with one another and the Reviewing Editor has drafted this decision to help you prepare a revised submission.

Summary:

This is a clear, sharp, succinct manuscript aiming right at the heart of what might seem to be a minor issue in CFTR gating biophysics – the nature, in a gain-of-function mutant, of the very rare ATP-independent openings that the wildtype channel undergoes. But the issue is no longer a minor one, since the first successful CFTR drug – ivacaftor – works by enhancing these openings in certain CFTR mutant patients. The aim of this paper is in no way to find out how the drug actually works, but rather to understand the process that the drug works upon – the ATP-independent openings. Csanady's group applies their characteristically rigorous single-channel approaches to ask whether these spontaneous openings represent NBD conformational changes like the normal, well-studied changes occurring in wildtype ATP-driven gating, or if they behave differently. And the answer is clear, from the same sorts of double mutant cycle experiments used previously: the same kind of NBD closing – now without sandwiching ATP molecules – is coupled to channel opening. Moreover, the kinetics of the process reprise the NBD movements of wildtype, ATP-loaded, though understandably with less 'molecular glue' stabilization involved.

Overall, the manuscript is well-written and the conclusions are firm and very pertinent to the current frenzy of CFTR drug discovery underway.

Issues to address:

1) Figure 2. It appears that there is some significant rundown in the 'background' recording. The effect is subtle, but looks real. (But I can't tell because to the inherently anecdotal character of the displayed record.) Could this possibly undermine the quantification of energies being used here?

2) Figure 3 is somewhat confusing. At the bottom (just above -0.4 kT) there are to arrow pointing towards each other: why?

The y axis contains a break, which makes it difficult to find all the numbers back that are calculated in the manuscript. For example, the 3.3 kT (Discussion, fourth paragraph).

3) In the subsection “Mutant cycle analysis” it is stated that SEMs were calculated using the mean value for n. It is not clear why this was done. Also the justification is weird: there were "similar numbers of observations". Looking through the manuscript, the number of observation is not explicitly presented for all experiments; in addition, it varies from 5-16.

4) In the subsection “The site-2 NBD interface is already tightened in the transition state for spontaneous pore opening”: 0.62 +/- 0.34 kT is not significantly different from zero. Why not?

5) Figure 2 would benefit from labels indicating what exactly was analyzed.

6) In the first sentence of the subsection “Gain-of-function mutations allow quantitative characterization of spontaneous CFTR gating”: Replace 'instantaneously' by 'rapidly' or some such adverb.

7) The manuscript is also generally well written, with the exception of the first sentence of the Abstract, which is full of commas. Please rewrite.

---

## [Author Response]

*Issues to address:*

1) Figure 2. It appears that there is some significant rundown in the 'background' recording. The effect is subtle, but looks real. (But I can't tell because to the inherently anecdotal character of the displayed record.) Could this possibly undermine the quantification of energies being used here?

There are two features in the recordings that the Reviewers might have perceived as "rundown".

First, the current decline upon ATP removal is typically biexponential, which is better appreciated by looking at the macroscopic current trace in Figure 1. The rapid decline with a large fractional amplitude likely reflects immediate loss of ATP from low-affinity site 2, whereas the subsequent slower phase is due to slower loss of ATP from high-affinity site 1 (see Tsai et al., 2010; J Gen Physiol 135:399-414). The current typically settles to a quasi-steady level within ~1 min following ATP removal. To extract gating parameters of unliganded channels we therefore analyzed only the last ~4 min of the 5 min ATP-free segments, as indicated in Methods. Of course in single-channel recordings loss of ATP from site 1 is a discrete event and so the actual time required to reach this "unliganded" steady state is variable. Thus, in individual recordings the precise starting time point for the "steady-state" segment to be analyzed was chosen by eye. E.g., for the trace in Figure 2 only the last ~3 min were analyzed. We have added a corresponding statement to Materials and methods (subsection “Excised inside-out patch recording”, last paragraph).

Second, the currents in ATP appear slightly smaller in the rear bracketing segments, as apparent in the macroscopic recording in Figure 1. One reason for this is that following a prolonged exposure to an ATP-free solution not all channels recover instantaneously, giving rise to the slowly creeping phase of the current in the second bracket in Figure 1. This phenomenon is also well known, and much more pronounced for WT CFTR (see Figure 1), but the underlying mechanism is unclear (see Tsai et al., 2010; J Gen Physiol 135:399-414). However, both in our background construct, and in the single- and double-mutants studied in this background, the open probability in the rear bracket, following full "reactivation", was only slightly smaller than in the front brackets (see Figure 1—figure supplement 2). We do not think that small decline – probably due to dephosphorylation, and observed similarly for all four constructs in the mutant cycle – would have substantially distorted the extracted values of interaction energies.

*2) Figure 3 is somewhat confusing. At the bottom (just above -0.4 kT) there are to arrow pointing towards each other: why?*

The three double-headed arrows in Figure 3 are used in the manner of dimension lines in technical drawings, to gauge distances between open-state and closed-state values for the three free energy profiles. The 0.4 kT distance for the gray profile is too small to fit both arrowheads into it. We have therefore used two arrowheads pointing in opposite directions, consistent with technical drawing guide lines. It is true that these two arrowheads should not have been connected to each other – we have removed the connecting line.

The y axis contains a break, which makes it difficult to find all the numbers back that are calculated in the manuscript. For example, the 3.3 kT (Discussion, fourth paragraph).

Unfortunately, a plot in which all the cited numbers are readily visible cannot be constructed. This is not due to the y-axis break (which is necessary in order to be able to draw at least all the changes in free energies to scale, given that the absolute heights of the barriers are disproportionately large (and also uncertain)), but to the fact that energy levels for three states (closed, transition, open) are compared to each other under three conditions (WT unliganded, BG unliganded, WT ATP-bound). For such comparisons to be made, one of the states must be arbitrarily chosen as a reference, and we have chosen state C as our reference state. This choice allows changes in △G^o^_(T-C)_ and △G^o^_(O-C)_ – caused by the BG mutations (downward black arrows) or by ATP binding (downward gray arrows) – to be directly visualized, whereas changes in △G^o^_(T-O)_ can be inferred as the difference between the former two numbers but cannot be simply illustrated. A similar limitation arises regardless of the choice of the reference state.

3) In the subsection “Mutant cycle analysis” it is stated that SEMs were calculated using the mean value for n. It is not clear why this was done. Also the justification is weird: there were "similar numbers of observations". Looking through the manuscript, the number of observation is not explicitly presented for all experiments; in addition, it varies from 5-16.

The numbers of observations were included in parentheses for all experiments, in the form "0.11±0.04 (6)". We realize that these numbers might have been mistaken as citations to references, so we have now explicitly added "*n*=" throughout the text (e.g., "0.11±0.04 (n=6)" (subsection “Gain-of-function mutations allow quantitative characterization of spontaneous CFTR gating”).

The statement (subsection “Mutant cycle analysis”) on using the mean value of *n* for SEM calculations applies only for the △△G values calculated from the mutant cycles (but, cf. Statistics section for all other data). In fact, estimation of △△G standard errors is not trivial, so we have added a more detailed description of this mathematical procedure (in the aforementioned subsection). Of note, *n* values for the four corners of each mutant cycle were quite similar to each other: *n*=19-21 for Figure 2, and *n*=15-19 for Figure 2 (a few patches that were not sufficiently "clean" were analyzed for P_o_, but not for gating kinetics).

4) In the subsection “The site-2 NBD interface is already tightened in the transition state for spontaneous pore opening”: 0.62 +/- 0.34 kT is not significantly different from zero. Why not?

As stated in the Statistics section, differences were considered significant for p<0.05. For the above parameter p=0.09 (see Figure 2).

5) Figure 2 would benefit from labels indicating what exactly was analyzed.

Done.

6) In the first sentence of the subsection “Gain-of-function mutations allow quantitative characterization of spontaneous CFTR gating”: Replace 'instantaneously' by 'rapidly' or some such adverb.

Done.

*7) The manuscript is also generally well written, with the exception of the first sentence of the Abstract, which is full of commas. Please rewrite.*

Done: "In CFTR, the chloride channel mutated in cystic fibrosis (CF) patients, ATP-binding-induced dimerization of two cytosolic nucleotide binding domains (NBDs) opens the pore, and dimer disruption following ATP hydrolysis closes it."